# Gold-Coated Superparamagnetic Iron Oxide Nanoparticles Functionalized to EGF and Ce6 Complexes for Breast Cancer Diagnoses and Therapy

**DOI:** 10.3390/pharmaceutics15010100

**Published:** 2022-12-28

**Authors:** Marcela Cândido, Paula Vieira, Andrea Campos, Cristina Soares, Leandro Raniero

**Affiliations:** 1Nanosensors Laboratory, Research & Development Institute, University of Paraíba Valley, Av. Shishima Hifumi, 2911, Urbanova, São José dos Campos 12244-000, São Paulo, Brazil; 2Aix-Marseille University, FSCM (FR1739), CP2M, 13397 Marseille, France; 3Cellular Compartment Dynamics Laboratory, Research & Development Institute, University of Paraíba Valley, Av. Shishima Hifumi, 2911, Urbanova, São José dos Campos 12244-000, São Paulo, Brazil

**Keywords:** magnetic nanoparticles, core–shell nanoparticle, theranostic nanoprobe, photothermal therapy

## Abstract

Superparamagnetic iron oxide nanoparticles (SPIONs) have some limitations in the physiological environment, however, a modification on their surface, such as a core–shell structure with gold (SPIONs@Au), can enhance their applicability. In this study, SPIONs were synthesized by the chemical coprecipitation method, stabilized by sodium citrate, and followed by the gold-coating process. SPIONs@Au were functionalized with EGF-α-lipoic acid and chlorin e6 (Ce6)-cysteamine complexes, composing a Theranostic Nanoprobe (TP). The outcomes showed that the SPIONs@Au had changed in color to red and had an absorption band centered at 530 nm. The coating was verified in the TEM micrographs in bright and dark fields by EDS mapping, which indicated the presence of Au and Fe. The Ce6-cysteamine complex had a resonant band at 670 nm that enabled the diagnosis of biological samples using fluorescence analysis. In the measure of TNBC cell uptake, the maximum value of TP fluorescence intensity was obtained within 4 h of internalization. At 2 h, the incorporation of the TP in the cytoplasm as well as in the nuclei was observed, suggesting that it could be employed as a diagnostic marker. The PTT results showed significant percentages of apoptosis in the TNBC cell line, which confirms the efficacy of the TP.

## 1. Introduction

The conventional systems for drug delivery have improved due to the application of nanotechnology in the healthcare area. The morphology and size of nanoparticles are important parameters for their production and application, since an increased surface area increases reactivity and the release of ions, whilst the effectiveness of these particles depends on their composition and solubility [1]. Thus, the physical and chemical properties of nanoparticles are directly influenced by their shape, size, crystal structure, chemical composition, and dispersibility [2,3], which all affect the performance of the function of these materials in their applications [4]. Among the materials, iron oxide nanoparticles (IONPs) have unique magnetic properties such as having high magnetic susceptibility, being superparamagnetic, and having low Curie temperature [2,3].

The IONPs are inorganic particles that are composed mostly of magnetite (Fe_3_O_4_), maghemite (γ-Fe_2_O_3_), or hematite (α-Fe_2_O_3_). These nanoparticles have been used in biomedical applications for drug separation, drug delivery, hyperthermia, biosensors, and Photothermal Therapy (PTT). Due to their ability to increase the reactivity of drugs in combination therapies or as hyperthermia agents, their therapeutic applicability has emerged primarily for the treatment of cancer [1,5,6,7]. These nanoparticles in combination with conventional gadolinium compounds might be used in magnetic resonance imaging, acting as specific agents for T2 relaxation times in image generation [8]. They also can be directed to a specific region using an external magnetic field while they circulate through the bloodstream [9].

As synthesized systems, IONPs have some limitations, including fast agglomeration, oxidation in the tumor environment due to the high volume/superficial area ratio, chemical reactivity, and high superficial energy, which cause the loss of magnetism. Therefore, it is necessary to change the surface of these colloid solutions to make them biocompatible. For instance, the core–shell structure (a nucleus enclosed in a shell) provides an inert surface while also making it easier for the part to function, improving the stability and biocompatibility of these solutions [4,10].

Nanoparticle core–shells, also known as hybrid nanoparticles, with a ferrous core and a gold shell (SPIONs@Au), are multifunctional and improve the physical, chemical, and optical properties of both materials [11]. These nanoparticles are promising candidates for magnetic hyperthermia due to the magnetic properties of iron, which may cause thermal amplification in cancerous cells. In this process, the presence of gold shields iron from oxidation. In addition, the gold shell tends to reduce magnetic interactions between particles, improving biological medium dispersion [12]. They can also be used in PTT, as well as in molecule detection [11,13,14,15].

Phototherapy corresponds to a non-invasive therapeutic approach that is activated by light and uses photoactivated intermediate agents, such as photothermal or photosensitizer transducers and light sources, in NIR and light-emitting diodes (LEDs). NIR is one of the light sources frequently used to activate the therapeutic effect in oncological phototherapy, due to its depth of tissue penetration and its non-harmful absorption by water molecules and hemoglobin in biological tissues [16]. PTT involves the conversion of incident light into heat with the aid of photothermal transducer agents to facilitate thermal ablation in tumor tissues [15,16,17,18]. Significant advantages of PTT over conventional treatments include reduced invasiveness, high cancer cell specificity, and fast recovery [15,18,19].

In this context, the present work proposes the synthesis of a Theranostic Nanoprobe (TP), increasing specificity for cancer cells with overexpression of the epidermal growth factor receptor (EGFR), such as the MDA-MB-468 cell line. In this process, the synthesis of iron oxide nanoparticles was performed by the chemical coprecipitation method; they were stabilized with sodium citrate and the surface gold coating process was also performed (SPIONs@Au). The functionalization of SPIONs@Au was prepared by coating the surface with EGF-α-lipoic acid and chlorin e6 (Ce6)-cysteamine (photosensitizer) complexes, forming the TP. Furthermore, these nanoparticles were characterized by UV-Visible Spectroscopy, Dynamic Light Scattering, and Transmission Electron Microscopy (TEM). In addition to the material characterization, the SPIONs, SPIONs@Au, and TPs were used for Photothermal Therapy (PTT) with an LED at 808 nm in the MDA-MB-468 cell line. Through Flow Cytometry, it was possible to verify the predominant apoptosis type of death after treatment with SPIONs, SPIONs@Au, and TPs.

## 2. Materials and Methods

Iron (II) chloride tetrahydrate 99% (FeCl_2_·4H_2_O), iron (III) chloride hexahydrate 97% (FeCl_3_·6H_2_O), sodium citrate dihydrate (Na_3_C_6_H_5_O_7_·2H_2_O) solution 25% *w*/*v*, the iron standard for ICP, hydroxylamine hydrochloride 99% (NH_2_OH·HCl), sodium acetate >99% (CH_3_COONa), 1,10-phenanthroline monohydrate >99%, ammonium hydroxide (NH_4_OH) 29% *v*/*v* and nitric acid (HNO_3_), Sulfo-NHS, EDC, α-lipoic acid, EGF, cysteamine, chlorin e6, and deionized water (18.2 Ω) were used throughout the experiments. All reagents were purchased from Sigma-Aldrich (Merck KGaA, Darmstadt, Germany), except for Ce6 (Frontier Scientific, Inc., Newark, NJ, USA).

### 2.1. Synthesis of SPIONs and SPIONs@Au

The coprecipitation approach was used to create SPIONs, as described by Mérida et al. (2015) [20] and Massart (1981) [21] (adapted). In ultrapure water that had previously been deoxygenated with nitrogen gas, iron chloride solutions II (FeCl_2_·4H_2_O) and III (FeCl_3_·6H_2_O) at 0.2 mol/L and 0.4 mol/L, respectively, were produced separately. Stirring was maintained throughout the synthesis at a temperature of 85 °C and a pH of 8 to 9. The precipitation of iron oxides occurred through the addition of 35 mL of ammonium hydroxide (NH_4_OH). This sample was resuspended in Na_3_C_6_H_5_O_7_·2H_2_O, taken to high-end ultrasound for 30 min, and centrifuged at 700 rcf for 10 min, followed by discarding the supernatant with a magnet.

The SPIONs@Au were synthesized according to Elbialy and collaborators (2019) (adapted) [22]. SPIONs at a concentration of 0.5 mg/mL were added to the 0.5 mmol/L solution of gold chloride (HAuCl_4_—anhydrous form) and stirred for 15 min. Later, the heating was stopped, and the solution was kept agitating until it reached room temperature (ELBIALY et al., 2019 [22]).

### 2.2. Synthesis of Theranostic Nanoprobe (TP)

Theranostic Nanoprobes (TPs) were produced according to the adaptation of the experimental procedure [23,24,25]. The Ce6-cysteamine complex was formed through carbodiimide chemistry, an amide bond reaction between the carboxylic groups present in the chlorin e6 molecule (Ce6) with cysteamine [23,24], which is used as a photosensitizer. The EGF-α-lipoic acid complex was formed by modifying the human EGF protein with the addition of α-lipoic acid [25]. The previously synthesized SPIONS@Au were functionalized to the complexes, EGF-α-lipoic acid and Ce6-cysteamine in 1:1 proportions, for full coverage of the colloid surface area. These complexes were linked to the core–shell nanoparticles by a gold–sulfur bond. First, SPIONS@Au were incubated with the EGF-α-lipoic acid complex under constant agitation for 24 h. After this period, the Ce6-cysteamine complex was added and the final solution was again kept under continuous stirring. After incubation, the TPs formed were purified by numerous centrifugations, quantified, and characterized.

### 2.3. Nanoparticles Characterization

The concentration of iron present in the SPIONs was determined by o-phenanthroline assays, and the result was also obtained by UV-Visible Spectroscopy. In this analysis, the SPIONs stabilized with Na_3_C_6_H_5_O_7_·2H_2_O were digested in 70% nitric acid (HNO_3_) for around 12 h in a dry bath at 101 °C. Subsequently, 10 µL of the digested sample was evaporated at 115 °C, followed by the addition of 46 µL of deionized water. To reduce Fe^3+^ ions to Fe^2+^, 30 µL of hydroxylamine hydrochloride (NH_2_OH. HCl) was added, and the reaction was kept at rest for 1 h. After reduction, 49 µL of sodium acetate (NaO_2_CCH_3_) was added as a buffering agent in addition to 75 µL of 1,10-phenanthroline monohydrate (C_12_H_8_N_2_·H_2_O), which formed the Fe(II)-orthophenanthroline complex. The absorbance of that complex was measured on a spectrophotometer, Synergy HT Multi-Detection Microplate Reader (BioTek Instruments, Winooski, VT, USA), at a wavelength of 508 nm.

The hydrodynamic size and Zeta potential were determined by Dynamic Light Scattering using the equipment ZetaSizer Nano—ZS90 (Malvern Panalytical, Malvern, UK). A volume of 200 μL of nanoparticles was added to the polystyrene cuvette (model ZEN0118, Sarstedt, Nümbrecht, Germany) to verify the particle size distribution. Three measurements were taken at a measuring angle of 90°, with a race number of 10 and a duration length of 30 s. Zeta potential analysis determined the stability and surface charge of the colloidal solutions when an electric potential differential was applied through a cuvette with gold electrodes. Using a syringe, the solutions were injected into capillary cuvettes with electrodes (model DTS1070, Malvern Panalytical, Malvern, UK). The mathematical model was defined by Smoluchowski, and an average of three analyses per sampling were performed to obtain the results. The nanoparticles were also characterized by UV-Visible Spectroscopy in a DeNovix DS-11 spectrophotometer in the 190–840 nm region, with a spectral resolution of 1 nm. In all readings, 2 μL of the sample was measured.

For TEM sample preparation, 10 µL of each sample was dried on copper grids coated with a 300-mesh carbon film, with a drying oven temperature of 40 °C. The morphology measurements were performed in the FEI TECNAI G20 LaB6 Transmission Electron Microscope (TEM) coupled with an Energy Dispersive X-ray (EDS) Oxford Silicon Drift Detector (Oxford Instruments, Abingdon, Oxon, UK). Bright- and dark-field TEM images as well as EDS spectra and maps were acquired at 200 kV.

### 2.4. Cell Culture

The human mammary adenocarcinoma cell line MDA-MB-468 was purchased from the Rio de Janeiro cell bank, which has the identification BCRJ code: 0166. The cells were cultivated in sterile 25 cm^2^ polypropylene bottles with L-15 culture medium (L4386-10x1L Leibovitz, Sigma Life Science, Merck KGaA, Darmstadt, Germany) and supplemented with 10% (*v*/*v*) Fetal Bovine Serum (16000-044, Life Technologies, Merck KGaA, Darmstadt, Germany). The culture conditions were kept at 37 °C and 95% humidity level (Water Jacket series 8000, Thermo Scientific, Marietta, GA, USA). After trypsinization with 0.25% trypsin-EDTA solution, the medium was changed every two days and the cells were subcultured every 15 days. All experiments with this cell line were performed in triplicate.

### 2.5. Confocal Microscopy Analysis

Confocal Fluorescence Microscopy analysis was performed to confirm the internalization of TPs in the MDA-MB-468 cell line. For the assays, 4 × 10^5^ cells/mL were added, and circular borosilicate coverslips positioned inside a 24-well plate were fixed. After 24 h of cell adhesion, the complete culture medium was replaced by TPs with a final concentration of 2 µg/mL of Ce6, with an incubation time of 2 h. After this period, the wells were washed several times with phosphate-buffered saline (1× PBS, pH 7.2). Coverslips were fixed with paraformaldehyde (4%), removed from wells, washed by immersion in PBS, and fixed on slides with Prolong Gold Antifade Mountant reagent with DAPI blue DNA marker [26]. Images were obtained using a Zeiss LSM 700 laser-scanning Confocal Fluorescence Microscope (Carl Zeiss AG, Oberkochen, Germany) with a Zeiss Plan-apochromat 63×/1.4 Oil Iris M27 objective. Images were obtained using a 405 nm laser line (5 mW) for DAPI (435 nm emission) and a 639 nm laser line (5 mW) for Ce6 (669 nm emission). Finally, the images were processed in the ZEISS ZEN 2 Microscope software, Version 10.0.19044.

### 2.6. Flow Cytometry Analysis

Flow Cytometry analysis was used to investigate the time of the TPs’ internalization in the MDA-MB-468 cell line through fluorescence intensity. The cells were plated with 1 × 10^5^ cells/mL, and after 24 h of cell adhesion, the culture medium was replaced by TPs with a final concentration of 2 µg/mL of Ce6 and incubated at times of 30, 60, 120, 180, 240, and 300 min. Subsequently, the cells were washed with phosphate-buffered saline solution (1× PBS, pH 7.2) and removed from the wells using 0.025% trypsin-EDTA solution. After removing the trypsin solution, the cells were transferred to sterile microtubes and washed with phosphate-buffered saline (PBS, pH 7.2). Data were collected by an Accuri™ C6 Plus Flow Cytometer (BD Life Sciences, San Jose, CA, USA) with 488 nm and 640 nm laser excitation. Forward (0° ± 13°) and side (90° ± 13°) light scattering detection were performed. Optical emission detection filters used were the following: FL1 533/30 nm (e.g., FITC/GFP); FL2 630/22 nm (e.g., PE/PI); FL3 682/23 nm (e.g., PerCP); and FL4 660/20 nm (e.g., APC). Finally, the collected data were processed in the BD Accuri C6 system software, Version 1.0.27.1.

Flow Cytometry analysis was performed to investigate the type of cell death after PTT, with SPIONs and SPIONs@Au at concentrations of 75 μg/mL and TPs at 2 μg/mL in the MDA-MB-468 cell line. All colloidal solutions were used for this analysis, which were incubated at 37 °C with 95% atmospheric air for 2 h. After incubation, cells were washed in phosphate-buffered saline (pH 7.2) and separated from wells using 0.025% trypsin-EDTA solution. Then, cells were transferred to sterile microtubes and received annexin V (AnnV) and propidium iodide (PI) staining for differentiation into viable (AnnV− PI−), apoptotic (AnnV+ PI−), and necrosis-like (AnnV+ PI+) cells. Cells were repeatedly washed in phosphate-buffered saline after staining (pH 7.2). The Accuri^TM^ C6 Plus Flow Cytometer (BD Life Sciences, San Jose, CA, USA) was used to gather the data, which were then analyzed using the Accuri^TM^ C6 system software, Version 1.0.27.1.

### 2.7. In Vitro Photothermal Therapy

For the PTT assay, 4 × 10^5^ cells/mL (MDA-MB-468) was applied to 24-well plates containing a complete culture medium. After 24 h of assessment, the medium was replaced by TPs (2 µg/mL), SPIONs, and SPIONs@Au (75 µg/mL). The positive control group received an overdose of dimethylsulfoxide (DMSO), and the negative control group received only a new culture medium. The cells were incubated for 2 h under culture conditions. After the incorporation time, the medium containing colloidal solutions was removed, the wells were washed with PBS, and a new complete culture medium was added for irradiation using an LED at 808 nm 3.3 J/cm^2^ for 10 min.

## 3. Results

For the coating study of SPIONs, the concentrations of the colloidal solution and HAuCl_4_ were initially standardized. The best results found for the coating were for HAuCl_4_ at the concentration of 0.5 mmol/L, Na_3_C_6_H_5_O_7_·2H_2_O at a concentration of 62.5 mmol/L, and SPIONs at 0.5 mg/mL. The reaction time after the color change of the solution was 15 min, and the solution was continuously stirred until it cooled down to room temperature. Figure 1A shows the UV-visible spectra of the SPIONs and SPIONs@Au with a band centered at 530 nm as well as a supernatant sample, which was collected after the colloidal solution was precipitated by the application of an external magnetic field. Furthermore, Figure 1C shows the DLS in the particle number of SPIONs (0.5 mg/mL used as a seed for the core–shell nanoparticles grown), with a hydrodynamic diameter of 21 ± 7.80 nm, polydispersity index (PdI) of 0.20, and a Zeta potential value of −48.0 ± 6.68 mV (Figure 1B). Figure 1D shows the DLS of the SPIONs@Au, in particle number, with a diameter of 40.99 ± 13.34 nm, PdI of 0.29, and Zeta potential value of −54 ± 6.00 mV (Figure 1B), remaining in the stability region.

A bright-field TEM micrograph of SPIONs is shown in Figure 2, making it possible to confirm the spherical morphology of the nanoparticles produced by coprecipitation and stabilized by sodium citrate. After treatment and analysis using the ImageJ program, version 1.48, it was possible to calculate the mean diameter of SPIONs, considering at least 500 nanoparticles with a proportion between their dimensions ≤ 1.6. To obtain the histogram, also shown in Figure 2, the log-normal function was applied to the data and a value of 7.20 nm in diameter was obtained with a standard deviation of 0.12 nm.

Figure 3 shows the morphology, size distribution, and elementary characterizations of the SPIONs@Au. Figure 3A displays TEM micrographs of the SPIONs@Au, indicating approximately spherical nanoparticles. EDS spectra shown in Figure 3C,E were acquired from points 1 and 2 indicated in TEM images Figure 3B,D of the SPIONs@Au in bright and dark fields, respectively. Confirmation of core–shell nanoparticle formation occurred through EDS mapping, in which the presence of Fe (Figure 3C,G) and Au (Figure 3E,H) were identified.

The functionalization of SPIONs@Au with the Ce6-cysteamine and EGF-α-lipoic acid complexes resulted in the formation of TPs. Figure 4 shows the UV-Visible spectra, with the presence of resonant bands centered at 402 nm (Soret’s Band) and 670 nm (Q Band) referring to the FS. For the characterization and quantification of Ce6, the Q band corresponds to the region of interest.

MDA-MB-468 is a triple-negative breast cancer cell line (TNBC), which is an excellent candidate for an alternative treatment since it has the fewest treatment options and the worst prognosis. Figure 5 shows the internalization time of TPs in the MDA-MB-468 cell line, with incubation times of 30, 60, 120, 180, 240, and 300 min.

The internalization of TPs in the MDA-MB-468 cell line was also investigated by utilizing Confocal Microscopy analysis, as shown in Figure 6. The presence of TPs in the cytoplasm is marked by the fluorescence of the Ce6 complex, colored red. As a result, the combination of this fluorescence makes it possible to assess the incorporation of TPs and their location in the cell cytoplasm.

The efficiencies of SPIONs, SPIONs@Au, and TPs associated with PTT were analyzed in the treatment of the TNBC cell line. Thus, Flow Cytometry was used to determine the type of death of the MDA-MB-468 cell line registered after PTT treatment. Figure 7A–C show the cytotoxic effects of SPIONs, SPIONs@Au, and TPs with PTT on the TNBC cell line, determining the cell death type as viable (AnnV− PI−), apoptotic (AnnV+ PI−), or necrosis-like (AnnV+ PI+) cells. To establish the type of death, two control groups were necessary, a positive control (Figure 7D) and a negative control (Figure 7E). The negative control had 90.4% viable cells, which is acceptable due to cell manipulation. Figure 7F summarizes all the results in the graphic of bars, highlighting the control groups.

## 4. Discussion

The core–shell of the SPIONs@Au was synthesized using a SPION seed. The quantification of SPIONs is a critical step, but Fe can be quantified by the formation of the Fe (II)-orthophenanthroline complex through the absorbance maximum at 508 nm. Thus, a Fe_3_O_4_ concentration of 2.36 mg/mL was used in the synthesis process, and SPIONs@Au were formed using a HAuCl_4_ solution that was initially kept stirring and heated to boiling. Subsequently, SPIONs and sodium citrate were added to the HAuCl_4_ solution, the reaction occurred for 5 min, and the change in the solution coloration was observed [4,27].

After the SPIONs’ coating, the first evidence of core–shell nanoparticles was registered by the change in color, which is explained by the presence of a resonance band centered at 530 nm in the UV-Visible spectra (Figure 1A), concomitant to an increase in the hydrodynamic diameter (Figure 1C,D). The hydrodynamic diameter increased from 21 nm (SPIONs) to 41 nm (SPIONs@Au), suggesting the formation of a stable and colloidal core–shell with Zeta potential equal to −54 mV. Indeed, an absorption band centered in a range of 500–530 nm is characteristic of gold particles at the nanoscale [28,29,30], and the SPIONs@Au remain with a magnetic behavior due to the iron core. The confirmation of the non-formation of gold nanoparticles occurred through the exposure of these nanoparticles to an external magnetic field, in which the supernatant obtained was quantified, showing no absorption band in UV-Visible spectra (Figure 1A).

A crucial physical property used to adjust the magnetic properties and surface area of SPIONs is their diameter [7]. The TEM micrograph of SPIONs shown in Figure 2 confirmed the spherical morphology of the SPIONs stabilized by sodium citrate, with a physical diameter value of 7.2 nm (Figure 2) being statistically obtained. In Figure 3, the TEM micrographs of SPIONs@Au indicated the presence of approximately spherical nanoparticles, with EDS results that allow the chemical characterization of materials [31,32]. In the bright-field analysis, the element Fe is shown in gray structures, while the element Au is found in black. When the same area was performed in dark-field analysis, the Au is represented by the gray structures, and vice versa. These results were confirmed in the EDS analysis of Spectrum 1 and Spectrum 2, identifying mainly the chemical elements Fe and Au, respectively. The copper identified is due to the TEM grid used as a sample substrate. As the process for obtaining SPIONs@Au begins with stabilized and synthesized SPIONs, it was noticed that some nanoparticles had been coated and others had not. This result was confirmed by the EDS map of the Fe and Au distribution (Figure 3F–H).

Figure 4 demonstrated the production of TPs by functionalizing SPIONs@Au with the complexes Ce6-cysteamine and EGF-α-lipoic acid, due to the appearance of resonant bands centered at 402 nm (Soret’s Band) and 670 nm (Q Band), which are consistent with the formation of TPs [23,24,33]. Figure 5 shows the internalization time of TPs in the MDA-MB-468 cell line, indicating an increase in fluorescence intensity up to 240 min. However, at 120 min they already have significant values of fluorescence intensity. Apart from that, the Ce6 might be a useful imaging agent for monitoring the delivery and treatment via PDT [34].

Nevertheless, the majority of photosensitizers have limitations in clinical applications, such as low accumulation for the specific target cells, high hydrophobicity, and low solubility; therefore, their directed delivery becomes necessary [35]. Thus, Confocal Microscopy was used to visually show the internalization of TPs by cells (Figure 6), demonstrating the presence and effectiveness of the internalization of TPs in the cytoplasm and some nuclei in the MDA-MB-468 cell line. These images are obtained because the fluorescent dye DAPI (2-(4-Amidinophenyl)-6-indolecarbamidine dihydrochloride) stains the cell nucleus with blue staining due to the chemical affinity in the A-T region of the DNA, and TPs are labeled by the fluorescence of the Ce6 complex, colored red [26].

According to HERNÁNDEZ-HERNÁNDEZ et al. (2020), the surface charge of nanoparticles is directly related to their distribution throughout the body. Nanoparticles with neutral charges show little interaction with plasma proteins and have a longer residence time in the body. Anionic nanoparticles interact and internalize by adsorptive endocytosis, whereas those with a positive charge are quickly absorbed by electrostatic attractions, as the cell membrane has a negative charge [1]. As is possible to observe in Figure 6, the TPs showed internalization in the cytoplasm and the nucleus of some cells, being confirmed through the image of the TPs’ sample section. These results show the benefit of TPs for cell monitoring as well as an agent of Photothermal Therapy, mainly for TNBC, which has the overexpression of the epidermal growth factor receptor and the worst prognosis.

Indeed, the standard care for cancer treatment typically entails surgical resection, followed by the delivery of chemotherapy and radiation therapy, both harmful to healthy tissue. In contrast to PTT, near-infrared (NIR) light permits non-invasively penetrating tissues deeply, energizing phototherapeutic compounds exposed to photons to raise the temperature and causing cell death. Furthermore, the thermal ablation of healthy tissue is reduced substantially in PTT [15,16]. According to Tian et al., temperatures above 50 °C are required for the induction of apoptosis or necrosis in tumor cells. As a therapeutic result, damages to organs and tissue are observed. As an alternative, a PTT approach has recently been developed with temperatures below 45 °C [36].

In PTT, the type of death that is produced can either be necrosis or apoptosis, where laser power plays an important role. High laser power increases the probability of necrosis, but low power might benefit the apoptosis pathways in the cell [37]. However, cell damage is associated with specific morphological changes that determine the mechanism of the cell death pathway [38,39]. In this context, Figure 7 shows the type of death obtained after the PPT, revealing a prevalence of apoptosis. This finding is crucial, since studies were carried out in triplicate and the Flow Cytometry technology can analyze each cell separately, which cumulates in three hundred thousand events. In addition, the predominance of apoptosis in treatment is a good strategy, achieving ideal results since cell death occurs in a programmed and milder way, while in necrosis there is a lack of control with dissemination to the rest of the body.

## 5. Conclusions

The DLS and TEM analyses verified that SPIONs and SPIONs@Au were successfully synthesized and stabilized, with diameters less than 100 nm. The coating process of SPIONs with gold was confirmed by the presence of the absorption band centered at 530 nm, by the increase in the hydrodynamic diameter, and by TEM detecting the presence of gold and iron by EDS mapping. The Ce6 and EGF complexes were functionalized on SPIONs@Au, revealing the presence of the resonant bands centered at 402 nm (Soret’s Band) and 670 nm (Q Band). The TPs’ uptake by MDA-MB-468 cells showed a continuous increase up to 240 min, demonstrating the presence and effectiveness of internalization of TPs in the cytoplasm as well as cell nuclei. PTT treatment showed significant percentages of death by apoptosis in TNBC cells, which is a good achievement for this therapy with this potential application.

## Figures and Tables

**Figure 1 pharmaceutics-15-00100-f001:**
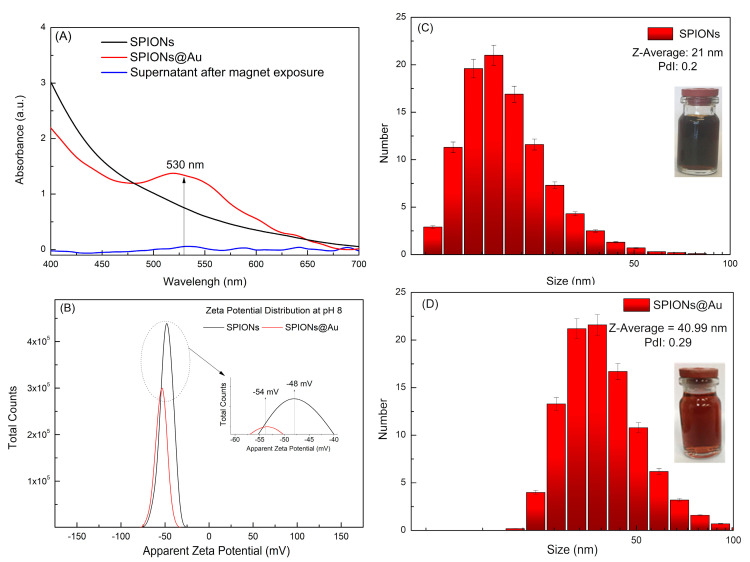
Characterization: (**A**) absorption spectra in the UV-Visible region, (**B**) Zeta Potencial Value for both nanoparticles, (**C**) hydrodynamic diameter distribution of SPIONs and (**D**) hydrodynamic diameter distribution of SPIONs@Au.

**Figure 2 pharmaceutics-15-00100-f002:**
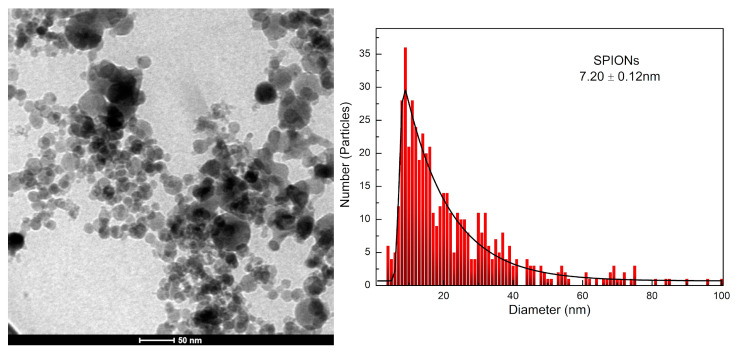
TEM micrograph of SPIONs and size histogram.

**Figure 3 pharmaceutics-15-00100-f003:**
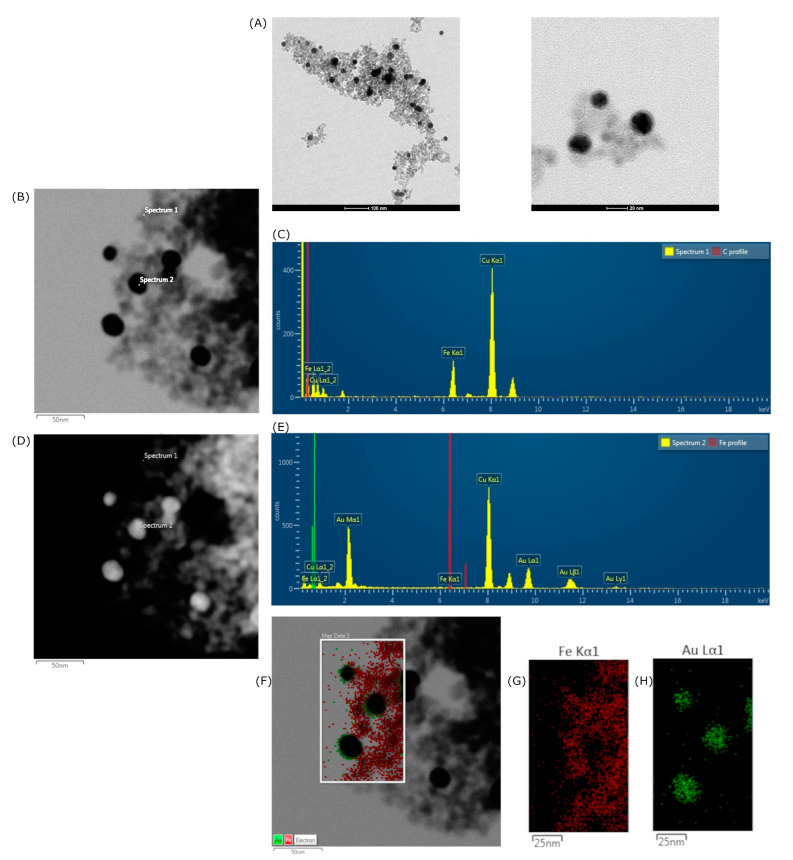
TEM and EDS analysis of SPIONs@Au: (**A**) SPIONs@Au; (**B**) bright-field micrograph; (**C**) EDS spectrum marker “Spectrum 1”; (**D**) dark-field micrograph; (**E**) EDS spectrum marker “Spectrum 2”; (**F**) micrograph with EDS map of Fe and Au distribution in the rectangle selected in the image; (**G**) Fe; (**H**) Au.

**Figure 4 pharmaceutics-15-00100-f004:**
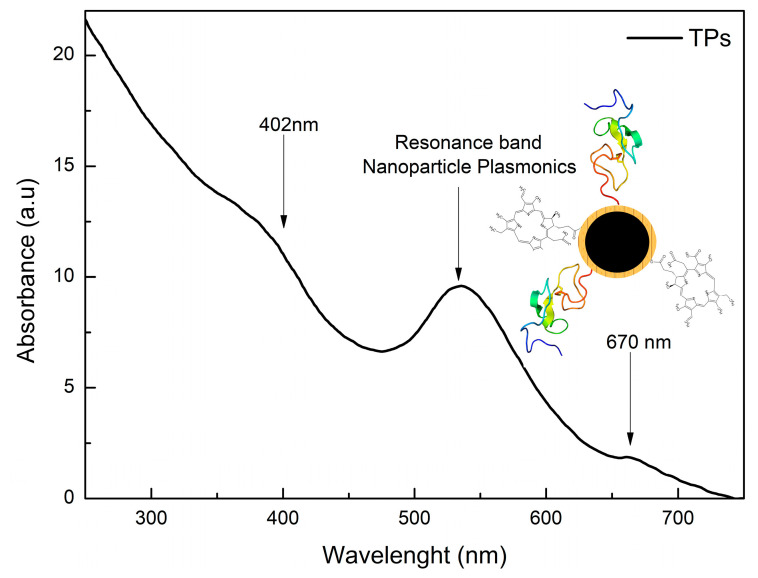
UV-visible spectrum of TPs and schematic representation.

**Figure 5 pharmaceutics-15-00100-f005:**
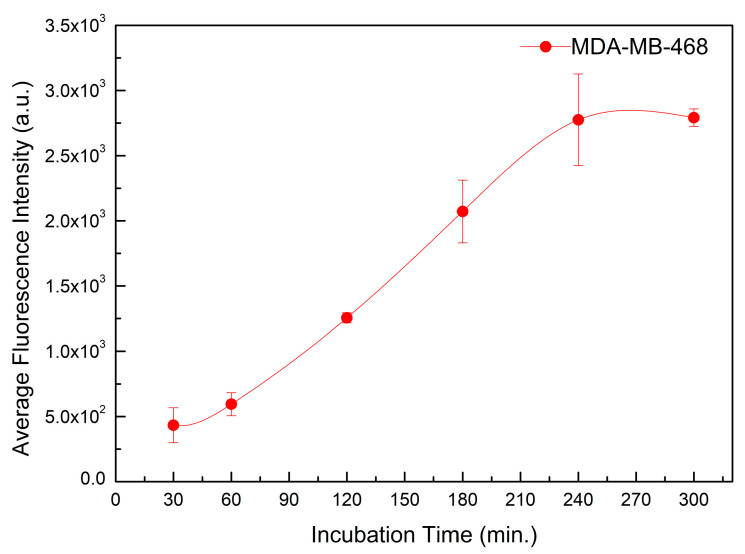
Analysis of the time of incorporation of TPs in the MDA-MB-468 cell line, with incubation times of 30, 60, 120, 180, 240, and 300 min.

**Figure 6 pharmaceutics-15-00100-f006:**
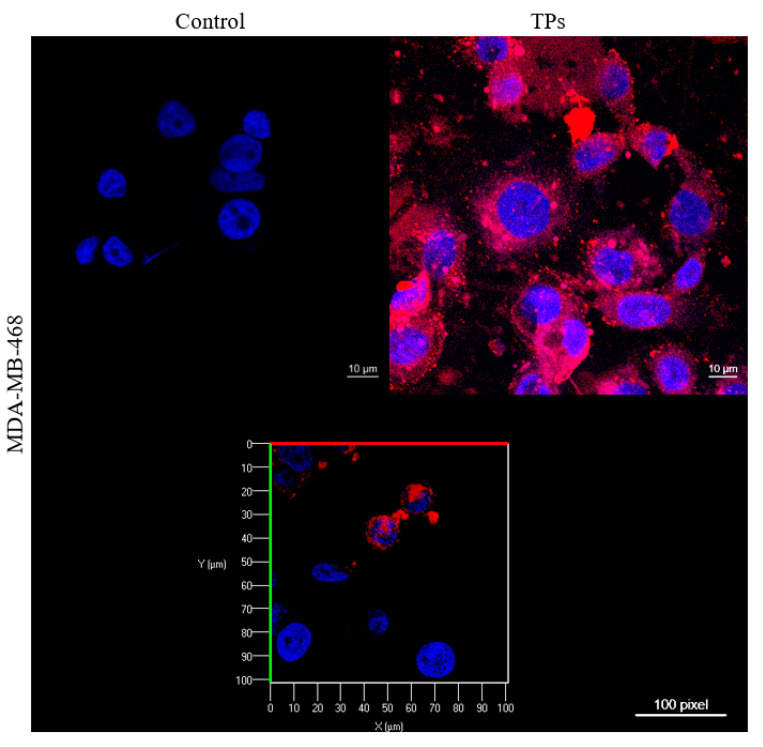
Micrograph of the MDA-MB-468 cell line for the analysis of the internalization of TPs. Blue staining reveals the fluorescence of DNA-bound DAPI in the cell nucleus and red staining refers to the fluorescence of TPs in the cell cytoplasm.

**Figure 7 pharmaceutics-15-00100-f007:**
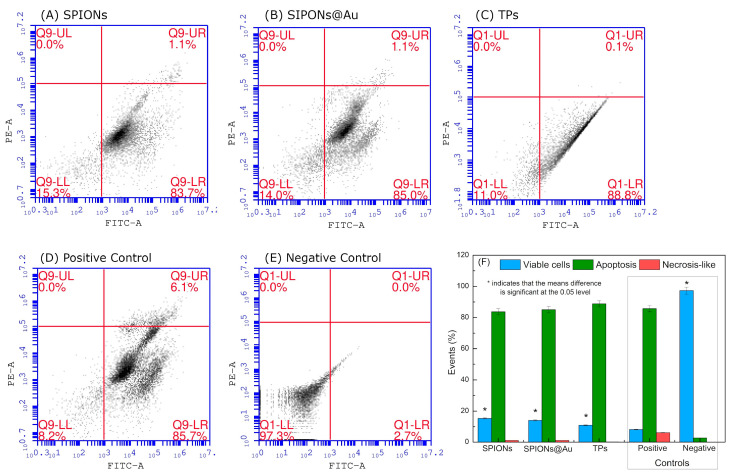
Type of cell death obtained with SPIONs (**A**), SPIONs@Au (**B**), and TPs (**C**) after PTT in the MDA-MB-468 cell line; (**D**) positive control and (**E**) negative control. (**F**) Bar graph with the result of the number of viable cells, necrosis, and apoptosis, obtained by Flow Cytometry for the control and nanoparticle groups.

## Data Availability

All data relevant to the publication are included.

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
