# Peer review of "Gold-Coated Superparamagnetic Iron Oxide Nanoparticles Functionalized to EGF and Ce6 Complexes for Breast Cancer Diagnoses and Therapy"

_pharmaceutics, 2022, doi:10.3390/pharmaceutics15010100_

Round 1

Reviewer 1 Report

In the “Gold-coated superparamagnetic iron oxide nanoparticles functionalized to EGF and Ce6 complexes for breast cancer diagnoses and therapy” paper, the authors synthesize a theranostic nanoprobe based on core-shell iron oxide-gold nanoparticles, for breast cancer diagnosis and therapy.

This is a well-written paper, on a topic of interest to the readers of the Pharmaceutics journal and can be considered for publication after a major revision.

My comments on this paper are as follows:

1. The photothermal agent investigated in this study is based on core-shell SPION-Au nanoparticles. However, the author did not show the unique advantage of their materials and they do not explain clearly the PTT effect. It is based on light-to-heat conversion, is it based on magnetic hyperthermia or a combination of both?

2. How is temperature increasing in the colloidal solution under LED irradiation for 10 minutes? Could the authors calculate the photothermal energy conversion efficiency for this material?

3. Since the TP resonance is centered at 530 nm, why the authors selected an 808 nm LED for irradiation? According to the literature the efficiency of PTT is higher when the NPs’ LSPR band is in resonance with the irradiation laser/LED?

4. The authors mentioned that only some of the SPIONs had been coated by Au. Is this affecting their stability in physiological media?  

5. Why is the TPs fluorescence intensity decreasing after 240 min incubation in the cells?

6. The quality/resolution of fig 1 and fig 2B should be increased

Author Response

The answer is attached

Reviewer 2 Report

In this work, Cândido et. al developed superparamagnetic iron oxide nanoparticles coated with gold for the diagnosis and therapy of breast cancer. The theme is interesting and timely, since breast cancer is one of the most prevalent type of cancers in the world. The developed nanosystems were physiochemically characterized using adequate methodologies. Their therapeutic potential was evaluated in vitro using human mammary adenocarcinoma cells. The presented study has scientific merit, and the manuscript is well structured, but some points should be addressed to improve the manuscript before publication. Below the authors can find some suggestions and questions:

Keywords: please avoid repeating words from the title, such as iron oxide nanoparticles.

The introduction could be improved by providing a brief summary of the recent advances of using SPIONs for breast cancer diagnosis and therapy, while justifying the novelty of this work

Also, the choice of EGF-α-lipoic acid and Ce6-cysteamine for the functionalization of the nanoparticles should be clearly justified in the introduction (or at least in some other part of the manuscript).

Line 113-116: How was achieved the conjugation of the nanoparticles with the EGF-α-lipoic acid and Ce6-cysteamine complexes? By a chemical reaction or by adsorption? Please clarify 

Lines 130-140: Experimental details for size and zeta could be shortened. 

Line 205: why the positive control group received an overdose or DMSO?

Lines 219-224: please present size, zeta potential and PDI results as mean and standard deviation (of at least three independent replicas)

Author Response

The answer is attached.

Reviewer 3 Report

Comments:

1.      From line 219 to 223, when the author mentioned the size, PDI and zeta potential, I recommend author add the standard deviation of those data.

2.      Any clearer TEM images of the SPIONs in Figure 2A? Most of the particles are aggregate together.

3.      In Figure 6, I recommend author stain the cell membrane as well to make sure the particles are internalized inside the cells.

4.      In Figure 7, I recommend the author add the statistical analysis when you report this data.

5.      In Figure 7, when you analyze the cell apoptosis, the four quadrant you drew on different values. For example, in Figure 7A, FITC-A line is between 10^2 to 10^3, but for Figure 7B, FITC-A line is on 10^3. Any explanation for this? I assume they should be on the same value and then you can compare the cell apoptosis level.

6.      Why can PTT cause cell apoptosis? Any explanation of that?

Author Response

The answer is attached.

Round 2

Reviewer 1 Report

The manuscript is improved after revision, therefore I suggest the publication

Author Response

English language and style were revised.

We appreciate your contributions to this work. I greatly appreciate your time and efforts. Thank you very much.

Reviewer 2 Report

The authors revised the manuscript and improved its quality. However, some points still need to be addressed before publication:

-In the introduction the novelty of this work should be clearly indicated, since gold coated SPIONs have been extensively reported for cancer photothermal therapy.

-  it should be clearly indicated in the manuscript that Ce6-cysteamine is used as a photosensitizer

- the authors should add to the materials the information that the EGF-α-lipoic acid and Ce6-cysteamine complexes are linked to the NPs by a gold–sulfur bond

Author Response

(The authors gave the same response as above.)

Reviewer 3 Report

All the comments are addressed properly. 

Author Response

(The authors gave the same response as above.)
